# Viro-Immunological Efficacy and Safety of Bictegravir/Emtricitabine/Tenofovir Alafenamide among Women Living with HIV: A 96-Week Post-Switch Analysis from the Real-Life SHiNe-SHiC Cohort

**DOI:** 10.3390/biomedicines12102311

**Published:** 2024-10-11

**Authors:** Agnese Colpani, Andrea De Vito, Andrea Marino, Manuela Ceccarelli, Benedetto Maurizio Celesia, Giuseppe Nicolò Conti, Serena Spampinato, Giulia Moi, Emmanuele Venanzi Rullo, Giovanni Francesco Pellicanò, Sonia Agata Sofia, Grazia Pantò, Carmelo Iacobello, Chiara Maria Frasca, Arturo Montineri, Antonio Albanese, Goffredo Angioni, Bruno Cacopardo, Giordano Madeddu, Giuseppe Nunnari

**Affiliations:** 1Unit of Infectious Diseases, Department of Medicine, Surgery and Pharmacy, University of Sassari, 07100 Sassari, Italy; colpani.agnese@gmail.com (A.C.); andreadevitoaho@gmail.com (A.D.V.); giuliamoi95@gmail.com (G.M.); giordano.madeddu@uniss.it (G.M.); 2Unit of Infectious Diseases, ARNAS Garibaldi Hospital, Department of Clinical and Experimental Medicine, University of Catania, 95122 Catania, Italy; bmcelesia@gmail.com (B.M.C.); giuseppecontichimica@gmail.com (G.N.C.); serenaspampinato93@gmail.com (S.S.); cacopard@unict.it (B.C.); giuseppe.nunnari1@unict.it (G.N.); 3Unit of Infectious Diseases, School of Medicine and Surgery, “Kore” University of Enna, 94100 Enna, Italy; manuela.ceccarelli@unikore.it; 4Unit of Infectious Diseases, G. Martino University Hospital, Department of Clinical and Experimental Medicine, University of Messina, 98122 Messina, Italy; emmanuele.venanzirullo@unime.it (E.V.R.); giovanni.pellicano@unime.it (G.F.P.); 5Unit of Infectious Diseases, AOE “Cannizzaro”, 95126 Catania, Italy; sonia.sofia@email.it (S.A.S.); graziapanto@tiscali.it (G.P.); carmelo.iacobello@gmail.com (C.I.); 6Unit of Infectious Diseases, “G. Rodolico-S. Marco” University Hospital, 95123 Catania, Italy; m.chiarafrasca@gmail.com (C.M.F.); a.montineri@libero.it (A.M.); 7Unit of Infectious Diseases, “Papardo” Hospital, 98158 Messina, Italy; antonioalbanese@icloud.com; 8Unit of Infectious Diseases, Ospedale Santissima Trinità, ASL8, 09121 Cagliari, Italy; goffredoangioni@asl8cagliari.it

**Keywords:** HIV, bictegravir, AIDS, WLWH, women and HIV, TAF, ART, integrase inhibitors

## Abstract

**Background/Objectives**: Out of 39.9 million adults living with HIV in 2022, 20 million were women. Despite bearing a significant burden, women remain underrepresented in clinical trials, including those for antiretroviral treatments (ART). This study evaluates the safety and efficacy of the bictegravir/emtricitabine/tenofovir alafenamide (B/F/TAF) regimen in a real-life cohort of 99 women with HIV (females with HIV, FWH) over 48 and 96 weeks. **Methods**: A retrospective cohort study utilized data from the Sardinian HIV Network and Sicilian HIV Cohort (SHiNe-SHiC) research group. The study included FWH, who started B/F/TAF as a treatment switch. The primary objectives were achieving and maintaining an HIV RNA level of <50 copies/mL at 48 and 96 weeks. Secondary objectives included treatment safety, durability, and reasons for discontinuation. Data on demographics, viro-immunological markers, lipid profiles, and treatment interruptions were extracted for analysis. **Results**: Among the 99 FWH, the median age was 51.9 years, and the median duration of HIV was 15.1 years. At baseline, 80.8% had undetectable HIV-RNA, which increased to 93.8% at 96 weeks. There was a statistically significant increase in CD4 cells/mL (48w *p* < 0.001, 96w *p* < 0.001) and CD4/CD8 ratio (48w *p* < 0.009, 96w *p* < 0.048), and reductions in total cholesterol (48w *p* < 0.003, 96w *p* < 0.006) and LDL (48w *p* < 0.004, 96w *p* < 0.009) levels at 48 and 96 weeks. Nine treatment interruptions were noted, with one due to adverse events. The regimen was well-tolerated overall. **Conclusions**: B/F/TAF demonstrated high efficacy and safety in this real-world cohort of FWH, highlighting the critical need for gender-focused research in HIV treatment. Ensuring equitable access to effective treatment options for women is imperative for the global health community’s efforts to eliminate HIV.

## 1. Introduction

Out of 39.9 million adults estimated to be living with HIV in 2022, 20 million were women. The World Health Organization reported that 540,000 women acquired HIV in the same year, and 230,000 died [1,2]. In 2024, a resolution focused on tackling gender-based discrimination in the context of the fight against HIV was adopted during the session of the Commission on the Status of Women. The Commission highlighted the disproportionate burden of the HIV pandemic on adult and adolescent women. Prioritizing women’s health should include addressing social determinants for gender disparities but also ensure more equitable access to diagnosis and treatment [3,4]. Health-seeking behaviors can differ across genders due to social determinants, disease burden, treatment-related adverse events, and pharmacokinetic and pharmacodynamic. Despite being long neglected, gender-driven differences in pathophysiology, pharmacokinetics, and pharmacodynamics are increasingly recognized as key factors for personalized care [5,6]. However, females, like other marginalized and neglected populations, have been repeatedly underrepresented in clinical trials, including in the field of HIV [7]. In a systematic review published in 2016, females nearly represented 20% of the antiretroviral treatment (ART) trial population, 38% in vaccine trials, and only 11% in curative strategies trials, with the lowest proportion included in publicly funded studies [7].

Among all completed trials on bictegravir/emtricitabine/tenofovir alafenamide (B/F/TAF) registered on clinicaltrials.gov, which have published demographic baselines, only two specifically address women; one enrolled 33 pregnant women in the second or third trimester [8]. The other included 234 virologically suppressed women switched to B/F/TAF; the results have already been published and showed the safety and efficacy of this regimen [9].

Regarding switch trials to B/F/TAF, women were 32%, 11%, 14%, and 0% in four different studies [10,11,12,13]; in a switch study among older people, women recruited were barely 13% [14].

In the ALLIANCE trial, assessing the safety and efficacy of B/F/TAF versus Dolutegravir + Emtricitabine/Tenofovir Disoproxil Fumarate in naive, Hepatitis B coinfected adults, women recruited were 7% and 2%, respectively [15]; in a similar trial addressing switch in coinfected individuals, women were only 14% [16]. Similar proportions were reported in a trial evaluating the safety and efficacy of B/F/TAF versus Abacavir/Lamivudine/F, in which 9% and 10% of women were recruited, respectively [17]. Finally, in the BASE trial conducted among people who inject drugs, women were 21% [18].

It is clear that women are constantly underrepresented in medical research, and the HIV field is no exception [19,20,21]. Every contribution to a better understanding of gender-specific issues in the field of HIV is crucial, especially for an ART regimen that is widely used in clinical practice both for naïve and experienced people with HIV. It is pushed from this rationale that we decided to evaluate the safety and efficacy of B/F/TAF in a cohort of 99 females with HIV (FWH) in a real-life setting at 48 and 96 weeks from the switch.

## 2. Materials and Methods

We conducted a retrospective cohort study utilizing data from the Sardinian HIV Network and Sicilian HIV Cohort (SHiNe-SHiC) research group. The SHiNe-SHiC project, initiated in 2019, collects comprehensive data from people with HIV (PWH) at multiple infectious disease centers in Sardinia and Sicily, Italy.

The study included all FWH enrolled in the SHiNe-SHiC database who started B/F/TAF as a treatment switch.

The primary objective was to assess the efficacy of the B/F/TAF regimen in experienced FWH, defined by achieving and maintaining an HIV RNA level of <50 copies/mL at 48 and 96 weeks. The secondary objectives included evaluating the treatment’s safety, the regimen’s durability, and the reasons for any treatment discontinuation.

### 2.1. Data Collection

Data were extracted from the SHiNe-SHiC database, including demographic details, risk factors for HIV acquisition, viro-immunological data (such as CD4 cells count and HIV RNA copies/mL), lipid profiles, creatinine, transaminases, blood glucose, and information regarding treatment interruptions. This data collection allowed for a detailed analysis of the characteristics and outcomes of individuals on the B/F/TAF regimen.

### 2.2. Statistical Analysis

Descriptive statistics summarized demographic and clinical characteristics. Quantitative variables were presented as medians with interquartile ranges (25th−75th percentiles) or mean with standard deviations (SD), depending on the distribution normality. Qualitative variables were described using absolute and relative frequencies (percentages). The Shapiro-Wilk test assessed the normality of quantitative data.

We analyzed data at three time points: baseline, 6 months (±1 month), and 12 months (±1 month). After assessing distribution normality, the student T-test and the Wilcoxon rank-sum test were used to identify significant changes across these time points. A *p*-value of <0.05 was considered statistically significant.

### 2.3. Ethical Considerations

The SHiNe-SHiC project adheres to ethical standards consistent with the Declaration of Helsinki. The relevant ethics committee approved the study. All participants provided written informed consent to partake in the study. The study was conducted in accordance with the Declaration of Helsinki and approved by the Provincial Ethics Committee of Messina (SHICohort-protocol code 34/17 of 22 March 2017, date of approval 22 May 2017).

Data collection and management were conducted strictly with privacy laws and regulations, including the European Union General Data Protection Regulation (GDPR). Patient data were anonymized and securely stored to ensure confidentiality and data integrity.

## 3. Results

We included 99 FWH in the study, with a median age of 51.9 years (IQR 43.1–57.8) and a duration of HIV infection of 15.1 years (IQR 5.1–24.7). CD4 T-cells nadir was 217 (IQR 57–360). Overall, five of them had positive HBsAg, and 15 had a history of HCV.

The most common comorbidity was dyslipidemia (38.4%), followed by hypertension (19.2%) and psychiatric disorders (17.5%) (Table 1).

### Efficacy

At the baseline, 80,8% of FWH had undetectable (HIV RNA < 50 copies/mL) HIV-RNA; six (6.1%) had a viral load between 50 and 200 copies/mL, and 13 (13.1%) over the 200 copies/mL. During follow-up, we observed an increase in the percentage of FWH with an undetectable viral load reaching 93.8% of the population (baseline level 80.8%), with only 6.2% of women with a detectable HIV-RNA at 96 weeks (Figure 1).

We noted a statistically significant increase in CD4 cells/mL and CD4/CD8 ratio at 48 and 96 weeks and no differences in CD8 cell counts (Figure 2).

Additionally, we noted statistically significant reductions in total cholesterol and low-density lipoprotein (LDL) at both 48 and 96 weeks, compared to baseline (Table 2). No significant changes were observed in transaminase levels, glucose levels, high-density lipoprotein (HDL), or triglycerides (Table 2).

We noticed an expected creatinine increase within the normal range at 48 weeks, stabilizing by 96 weeks. Finally, there were nine interruptions of B/F/TAF: three due to loss to follow-up, one death, one switch to long-acting treatment, one due to central nervous system (CNS) adverse events, and three cases due to patient’s choice.

## 4. Discussion

Our study addresses a critical gap in HIV research by focusing on the efficacy and safety of the B/F/TAF regimen in women, a group historically underrepresented in clinical trials despite bearing a significant burden of the HIV epidemic.

The findings from our retrospective cohort study suggest that switching to B/F/TAF is effective and safe for FWH. Particularly, in the present study, virological suppression significantly increased in FWH on B/F/TAF at 96 weeks. The virological efficacy of B/F/TAF is supported by abundant literature, making this regimen the first-line combination of choice in most naïve people and a safe and reliable option for experienced people with HIV. Data regarding women are scarcer; however, the existing literature is consistent with our findings. A study by Kityo et al. randomized 472 women to either switch to B/F/TAF or stay on a baseline regimen. HIV-RNA ≥ 50 cp/mL was reported in 1.7% of FWH on B/F/TAF at week 48; however, no interruption due to virological failure was reported [9]. A recently published trial conducted among pregnant virologically suppressed FWH confirmed the robust virological suppression maintained at 12 weeks post-partum, despite a slightly lower concentration of the three drugs during pregnancy than during post-partum follow-up. All 33 women in the trial maintained virological suppression up to a median of 16 weeks of follow-up post-partum [8]. A prospective study from an Italian cohort reported data from 241 FWH, of which 28 were treated naïve followed up for 12 months after the switch to B/F/TAF. The study reported a 100% efficacy in treatment-naïve FWH and 97% in treatment-experienced [22]. Similarly, the observational study by d’Arminio Monforte et al. reported data on B/F/TAF among naïve and experienced people with HIV; they included 167 naïve and 315 experienced FWH. The median follow-up of naïve participants was 69.8 weeks, with only 4.1% of virological failure reported, with no difference for gender. The median follow-up of the experienced was 146.3 weeks, with 2.5% virological failure; again, no difference by gender was reported [23].

Regarding immunological outcomes, CD4 cell count and CD4/CD8 ratio were augmented, underlining an amelioration of the immunological and inflammatory profile [24]. Regarding the virological outcome, this finding is consistent with what was reported from the BICSTaR cohort despite gender-specific outcomes not being reported [22]. The study by Kityo et al. did not report any statistically significant difference in CD4 cell count between B/F/TAF and comparator groups [9].

The amelioration of the immunological profile, reflecting lower levels of inflammation along with immune reconstitution, could be due to the virological efficacy of B/F/TAF compared to the previous regimens. In fact, we demonstrated that a higher number of FWH reached virological suppression at 96 weeks.

Regarding tolerability, the regimen was overall well tolerated, as described in the existing literature [25,26].

However, we observed a significant increase in creatinine levels at 48 weeks, which then stabilized by 96 weeks. This increase is compatible with the inhibition of the organic cation transporter 2 (OCT2) and MATE-1, a known effect of both bictegravir and dolutegravir [27,28]. OCT2 inhibition can lead to increased serum creatinine levels without necessarily indicating a true decline in renal function. Despite the increase in creatinine, no interruption due to kidney toxicity has been observed. Similar results have been observed in the BICSTaR cohort, which reported two discontinuations due to kidney impairment; however, no gender-specific details are given [22]. Of notice, the same study published as supplementary material that 50% of naïve and 48% of experienced FWH experienced at least one adverse event, statistically less than the male counterpart. Kityo et al. reported no differences in serum creatinine between the two groups. No mention of kidney toxicity is found in the other studies examined [9]. Nonetheless, creatine remained in the normal range, and the increase could be due to the switch from a non-STI-based regimen to an INSTI-based one.

Our cohort exhibited a range of clinical symptoms and comorbidities, with dyslipidemia being the most common (38.4%), followed by hypertension (19.2%) and psychiatric disorders (17.5%). The reduction in total cholesterol and LDL levels observed at both 48 and 96 weeks is clinically significant, particularly for individuals with dyslipidemia, as cardiovascular disease remains a major concern in people living with HIV [29,30,31]. The favorable impact of B/F/TAF on lipid profiles aligns with findings from previous trials, including the BICSTaR cohort, where similar reductions in lipid levels were noted among treatment-experienced individuals [18]; as for the other outcome, no gender-specific data are provided. Of interest, Kityo et al. did not find any significant difference between the comparator and B/F/TAF in this regard [9]. The reduction in total cholesterol and LDL may reduce cardiovascular risk, as already demonstrated by the REPRIVE trial [24].

In terms of psychiatric comorbidities, 17.5% of women in our cohort had pre-existing psychiatric conditions. One participant discontinued B/F/TAF due to central nervous system (CNS) side effects, a known but relatively uncommon adverse event. The impact of CNS side effects on individuals with psychiatric disorders should be carefully monitored, as noted in prior studies [32,33,34]. However, the low discontinuation rate in our cohort aligns with the existing literature, which reports good tolerability of B/F/TAF in most PLWH [8,9,22,23,35,36].

Our study is limited by its retrospective design, the lack of randomization and of a comparator group, and the small sample size. On the other hand, it provides real-life, gender-specific evidence, and a long follow-up characterizes it.

To our knowledge, this is the first observational study addressing the performance of B/F/TAF at 96 weeks in women, underscoring the worrying lack of gender-specific data in the HIV continuum of care.

## 5. Conclusions

Our study highlights the safety and efficacy of B/F/TAF in a real-world cohort of women living with HIV, underscoring the critical need for gender-specific research in HIV treatment. While previous studies have established the efficacy of B/F/TAF in mixed-gender cohorts, our findings offer valuable insights into its use among women, a population historically underrepresented in clinical trials. The data presented here confirm that B/F/TAF is a well-tolerated and effective regimen in achieving virological suppression and improving immunological outcomes in women. This study contributes to the growing body of evidence supporting the need for more gender-focused research to ensure that women have equitable access to the best treatment options available. Addressing the gender gap in HIV research is essential for improving outcomes and ensuring that treatment strategies are as inclusive as possible.

## Figures and Tables

**Figure 1 biomedicines-12-02311-f001:**
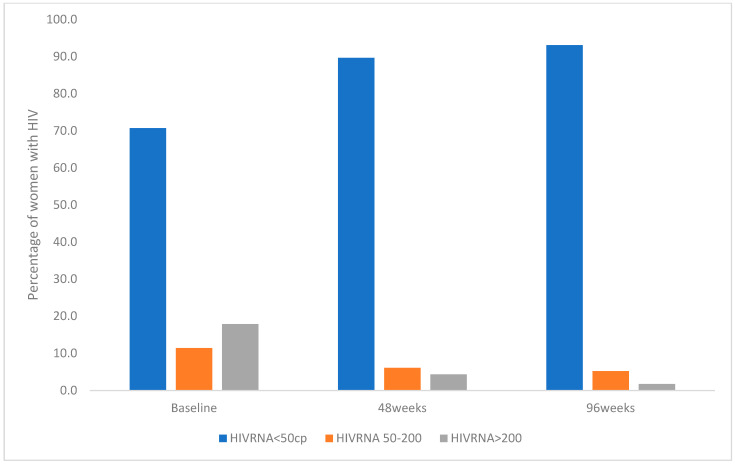
HIV-RNA in females living with HIV starting B/F/TAF at baseline, 48 weeks, and 96 weeks.

**Figure 2 biomedicines-12-02311-f002:**
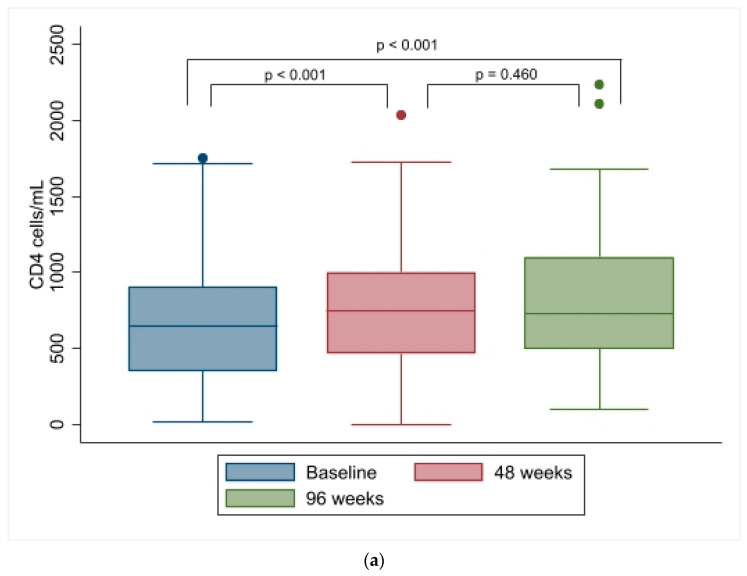
(**a**) CD4, (**b**) CD8, and (**c**) CD4/CD8 in 99 females with HIV starting B/F/TAF at baseline, 48 weeks, and 96 weeks.

**Table 1 biomedicines-12-02311-t001:** Demographical and clinical characteristics of 99 women living with HIV switching to TAF/FTC/BIC.

Characteristics	Participants
Age (years), median (IQR)	51.9 (43.1–57.8)
Duration of HIV (years), median (IQR)	15.1 (5.1–24.7)
HIV-RNA (copies/mL) Zenith, median (IQR) *	74,000 (15,000–350,000)
CD4 T-cells (cells/mL) Nadir, median (IQR) ^#^	217 (57–360)
Risk factors for HIV acquisition, n (%)	
Sexual, n (%)	89 (90.1)
IDU, n (%)	10 (9.9)
Smoking, n (%)	
Current	34 (34.3)
Former	3 (3.0)
Never	62 (62.7)
Daily alcohol intake, n (%)	15 (15.1)
History of AIDS-defining condition, n (%)	21 (21.2)
HBsAg positivity, n (%)	5 (5.1)
HCV antibodies, n (%)	15 (15.2)
**Comorbidities**	
Hypertension, n (%)	19 (19.2)
Dyslipidemia, n (%)	38 (38.4)
Cancer, n (%)	8 (8.1)
Diabetes, n (%)	3 (3.0)
Osteoporosis, n (%)	4 (4.1)
Psychiatric disorders, n (%)	17 (17.5)
**Not-ART treatments**	
Statins, n (%)	18 (18.2)
Antihypertensive, n (%)	16 (16.2)
Metformin, n (%)	2 (2.0)
Antidepressant, n (%)	14 (14.1)

* Data on 73 women; ^#^ data on 74 women; IQR: interquartile range.

**Table 2 biomedicines-12-02311-t002:** Results of biochemical exams at baseline, 48 weeks, and 96 weeks in 99 females with HIV treated with B/F/TAF.

	Baseline	48 Weeks			96 Weeks			
		Number of FWH	Median (IQR)	*p*-Value ^T0–T1^	Number of FWH	Median (IQR)	*p*-Value ^T0–T2^	*p*-Value ^T1.T2^
Creatinine mg/dL	0.75 (0.62–0.88)	91	0.79 (0.69–0.90)	0.019	58	0.80 (0.69–0.97)	<0.001	0.155
AST U/L	22 (18–29)	87	22 (18–29)	0.847	58	23 (19–28)	0.186	0.596
ALT U/L	20 (14–27.5)	87	21 (16–30)	0.116	58	21 (16–26)	0.375	0.779
Glycemia mg/dL	88 (82–98)	81	90 (80–97)	0.593	50	90 (83–100)	0.865	0.562
Total cholesterol, mg/dL	203 (168.5–237.5)	93	188 (163–219)	0.003	58	195 (164–217)	0.006	0.834
LDL, mg/dL	121.5 (99–149)	93	112 (92–142)	0.004	58	119 (88–135)	0.009	0.943
HDL	52.5 (45.5–63)	93	55 (44–65)	0.606	58	57.5 (49–66)	0.101	0.489
Triglycerides, mg/dL	98.5 (72.5–148.5)	93	95 (70–137)	0.095	58	94 (67–140)	0.097	0.475

T0 = baseline; T1 = 48 weeks; T2 = 96 weeks.

## Data Availability

The data presented in this study are available upon request from the corresponding author.

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
