# Peer review of "Viro-Immunological Efficacy and Safety of Bictegravir/Emtricitabine/Tenofovir Alafenamide among Women Living with HIV: A 96-Week Post-Switch Analysis from the Real-Life SHiNe-SHiC Cohort"

_biomedicines, 2024, doi:10.3390/biomedicines12102311_

Round 1

Reviewer 1 Report

Comments and Suggestions for Authors

The manuscript titled: Viro-immmunological efficacy and Safety of bictergravir/emtricitabine/tenofovir alafenamide among women living with HIV, a 96 weeks post switch analysis: Results from the real life SHINESHIC cohort" report a very interesting data mainly focuses on women living with HIV-1 and their conclusion states that these drug combination has high efficacy and safety in this real world cohort. However I have these burning comments with regards to the design of the study.

1. If this research was gender-gender focus research why the authors did not set up a cohort of males in the study so that they may compare the results?

2. In the introduction and abstract, HIV-1 statistics need to be updated since it now 2024 and UNAIDS report has different statistics.

3. In page 2 line 93. Why the authors argue that women are under represented in the medical research but most of the paper that reported here indicate that women were participating in the study? Can authors elaborate on that.

4. In the current study what was the comparison between men and women enrolled in the overall study?

5. Table 1 legend must on top and be descriptive. It is difficult the read the results presented in that table.

6. figure 1 legend must be improved and more descriptive and also note that there is error in baseline spelling.

7. All the figure legends must be improved and more descriptive on what the figure is showing.

8. table 2 legend must be on top.

9. In discussion, the study that are discussed are mostly from the women cohort and mostly comparison is the drug combination. There is no enough evidence that Men are the priority in medical research in order to address your hypothesis.

10. Your conclusion states that this the first study addressing the perfomance of B/F/TAF in women but there is no study that you referring where the same study performed in males.

My general opinion

It will be better to discuss the the safety and efficacy of drugs in women but not insisting that men are doing better while there is no comparator group in the study. Revise your conclusion

Author Response

  1. If this research was gender-gender focus research why the authors did not set up a cohort of males in the study so that they may compare the results?

AR: Thank you for this comment. Our study was intentionally designed to focus exclusively on women living with HIV (WLHIV) due to their historical underrepresentation in clinical trials, including those involving antiretroviral therapy. We believe that collecting and analyzing data from a female-only cohort is crucial to address this gap in evidence. Including a male cohort would have shifted the study’s focus and introduced variables that could obscure gender-specific findings. While there are existing studies focused on men or mixed populations, the strength of our study lies in its focus on women, providing real-world, gender-specific data.

  1. In the introduction and abstract, HIV-1 statistics need to be updated since it now 2024 and UNAIDS report has different statistics.

AR: Thank you for your comments. We have update the information

  1. In page 2 line 93. Why the authors argue that women are under represented in the medical research but most of the paper that reported here indicate that women were participating in the study? Can authors elaborate on that.

AR: We appreciate this opportunity to clarify. While women do participate in medical research, their inclusion is often in disproportionately small numbers. The studies we cited illustrate this: women represent a minority of participants, even in trials of antiretroviral therapies, where they account for a smaller percentage compared to men. For instance, in many switch trials of B/F/TAF, women constituted only 7% to 21% of participants. Therefore, despite their inclusion, women are still underrepresented relative to their disease burden, which reinforces the need for research specifically focusing on them.

  1. In the current study what was the comparison between men and women enrolled in the overall study?

AR: Our study was designed to focus on women exclusively, and there was no direct comparison with men. However, we acknowledge that comparisons between genders in the context of HIV treatment are valuable and have been reported in other studies. The objective here was to provide insights into the efficacy and safety of B/F/TAF specifically in women, a population that has historically been neglected in clinical research.

  1. Table 1 legend must on top and be descriptive. It is difficult the read the results presented in that table.

AR: Thank you for your comment. We fix it.

  1. figure 1 legend must be improved and more descriptive and also note that there is error in baseline spelling.

AR: Thank you for your comment. We fix it.

  1. All the figure legends must be improved and more descriptive on what the figure is showing.

AR: Thank you for your comment. We fix it.

  1. table 2 legend must be on top.

AR: Thank you for your comment. We fix it.

  1. In discussion, the study that are discussed are mostly from the women cohort and mostly comparison is the drug combination. There is no enough evidence that Men are the priority in medical research in order to address your hypothesis.

AR: We recognize this point. While our discussion primarily focuses on the data from women, there is substantial evidence showing that men have been overrepresented in HIV clinical trials. Our emphasis was on the underrepresentation of women rather than suggesting that men are prioritized per se. We will revise the wording to better reflect that the focus is on filling the gender gap in research rather than implying that men are exclusively prioritized.

  1. Your conclusion states that this the first study addressing the perfomance of B/F/TAF in women but there is no study that you referring where the same study performed in males.

AR: We agree that the conclusion could be clearer. While there are studies on the performance of B/F/TAF in mixed-gender cohorts, few provide gender-specific outcomes. Our study's significance lies in presenting real-life data specifically on women with HIV, which has been underreported in the existing literature. We will revise the conclusion to make it clear that while B/F/TAF has been widely studied in men and mixed cohorts, this is one of the first real-world studies focusing solely on women.

My general opinion

It will be better to discuss the the safety and efficacy of drugs in women but not insisting that men are doing better while there is no comparator group in the study. Revise your conclusion

AR: We appreciate this feedback. We will adjust the conclusion to emphasize the study's contribution to understanding the safety and efficacy of B/F/TAF in women, without framing it as a comparison to men. The revised conclusion will focus on the importance of providing gender-specific data and supporting equitable access to treatment options for women living with HIV.

Reviewer 2 Report

Comments and Suggestions for Authors

Dear Authors,

1. Although the design of the study is reasonable and the results deserve to be considered as interesting and impactful for the readership, the study is based on retrospective analysis. Therefore, it is necessary to accurately compare the results with those previously published in the literature and/or analysed in clinical trials, especially for the same drug regimens (B/F/TAF), and to reveal novel results obtained in your study.

2. Please discuss possible clinical symptoms and co-morbidities and/or complications associated with specific changes in clinical characteristics, including measured indicators.

Comments on the Quality of English Language

Moderate changes of English are needed.

Author Response

  1. Although the design of the study is reasonable and the results deserve to be considered as interesting and impactful for the readership, the study is based on retrospective analysis. Therefore, it is necessary to accurately compare the results with those previously published in the literature and/or analysed in clinical trials, especially for the same drug regimens (B/F/TAF), and to reveal novel results obtained in your study.

AR: Thank you for your insightful comment. We agree that it is crucial to compare our results with previously published literature to contextualize our findings. While there have been several studies on the efficacy and safety of B/F/TAF in mixed-gender cohorts, the number of studies specifically focusing on women is limited. We have highlighted key findings from clinical trials that involved both men and women, including the ALLIANCE, BASE, and BICSTaR cohorts, among others. These studies primarily involved a smaller percentage of women, and our study adds valuable real-world evidence by focusing exclusively on women. One of the novel findings in our study is the statistically significant increase in virological suppression (HIV-RNA <50 copies/mL) from 80.8% at baseline to 93.8% at 96 weeks, along with significant improvements in the CD4 count and CD4/CD8 ratio. Additionally, we observed clinically significant reductions in total cholesterol and LDL cholesterol, which are consistent with the results reported in trials among treatment-experienced people with HIV. However, our study specifically addresses these outcomes in women, contributing novel gender-specific data.

  1. Please discuss possible clinical symptoms and co-morbidities and/or complications associated with specific changes in clinical characteristics, including measured indicators.

AR: Thank you for this important point. We acknowledge that discussing the clinical symptoms, co-morbidities, and their associations with changes in clinical characteristics is essential for a more comprehensive analysis. In our study, the most common co-morbidities were dyslipidemia (38.4%), hypertension (19.2%), and psychiatric disorders (17.5%). As noted, we observed significant reductions in total cholesterol and LDL levels at 48 and 96 weeks. These improvements are particularly relevant for women with pre-existing dyslipidemia, as controlling lipid levels is critical for reducing cardiovascular risk in people with HIV.  Regarding psychiatric disorders, 17.5% of the women had pre-existing psychiatric comorbidities. One patient discontinued B/F/TAF due to central nervous system (CNS) side effects, which is a known but relatively uncommon adverse event associated with the regimen. We will expand the discussion to include the potential impact of CNS-related side effects on patients with pre-existing psychiatric conditions, drawing comparisons with other studies that have reported similar outcomes.

Round 2

Reviewer 1 Report

Comments and Suggestions for Authors

Thank you for addressing the comments. However the statistics of people living with HIV is now 39.9 million worldwide. please update that. In the abstract FWH must be written in full since it is the first mention.

I am satisfied with the comments.

Author Response

Thank you for addressing the comments. However the statistics of people living with HIV is now 39.9 million worldwide. please update that. In the abstract FWH must be written in full since it is the first mention.

Reply: We fixed what you suggested, thank you.